# Regulation/Non-Regulation/Dys-Regulation of Health Behavior, Psychological Reactance, and Health of University Undergraduate Students

**DOI:** 10.3390/ijerph18073793

**Published:** 2021-04-05

**Authors:** Mónica Pachón-Basallo, Jesús de la Fuente, María Carmen Gonzáles-Torres

**Affiliations:** 1Department of Theory and Methods of Evaluation and Educational and Psychological Intervention, Faculty of Education and Psychology, University of Navarra, 31009 Pamplona, Navarra, Spain; mpachonbasa@alumni.unav.es (M.P.-B.); mgonzalez@unav.es (M.C.G.-T.); 2Department of Psychology, Faculty of Psychology, University of Almeria, 04120 La Cañada, Almería, Spain

**Keywords:** SRL vs. ERL theory, self-regulation behavior, self/external regulation of health, self/external non-regulation of health, self/external dys-regulation of health, psychological reactance, student health

## Abstract

The Self-Regulation vs. External-Regulation Theory (2017) has postulated a continuum of regulation/non-regulation/dys-regulation that is present both in the individual and in the individual’s context. This gives rise to a behavioral heuristic that can predict and explain other health-related variables, such as psychological reactance and student health. On a voluntary basis, 269 university students completed validated questionnaires on variables of regulation, reactance and health. Using an ex post facto design, we performed correlational analysis and structural linear regression to build a structural equations model (SEM) with acceptable statistical values. The results showed various predicted relationships: self-regulation was associated with and positively predicted self-regulated health behavior; external health-regulating contexts were associated with and positively predicted self-regulated health behavior; non-regulatory and dysregulatory contexts negatively predicted self-regulated health behavior and students’ health itself, as well as positively predicting psychological reactance behavior. Implications are established for explaining variability in general and health-related self-regulation, as well as for intervening in these variables in health programs.

## 1. Introduction

University students’ health has become the object of research. Prior studies have revealed certain factors that affect student health, such as academic or money-related worries [1]. Other factors that show statistically significant association with students’ well-being are frequency of physical activity, current tobacco use, sleep quality, depression and the availability of mental health services [2,3]. The World Health Organization (WHO) [3] adds that development of skills to maintain interpersonal relationships, to face problematic situations, and to manage one’s emotions are essential for the health of this group of young people.

In this context, self-regulation (SR) behavior has been found to positively predict flourishing and student health, as well as to negatively predict procrastination [4]. Similarly, psychological reactance has been positively correlated with greater perceived loneliness and behavioral problems in adolescents and youths [5]. Therefore, the purpose of this research was to use correlations and a structural equations model (SEM) to establish whether the health-regulatory levels of the individual and of their context were linearly related to the individual’s general self-regulation, psychological reactance and health.

### 1.1. Self-Regulation Behavior (SR)

Lazarus and Folkman (1984) [6] conceptualized self-regulation as the set of processes and behaviors that support pursuit of personal goals within a changing external environment. Brown et al. (1999) [7] evaluated individuals’ behavioral self-regulation, considering it their ability to plan and manage their own behavior in a flexible way, according to desired outcomes.

Sufficient research evidence has established SR as a meta-ability [8,9] that it is associated with different personality variables, such as coping strategies [10]. It has also been found to positively predict health and flourishing [4]. Complementarily, self-regulation has shown a significant, negative association with poor sleep quality in an average university population, as well as negative predictive value of smartphone addiction [11]. Levels of self-regulation have also been found to correlate positively with degree of well-being [12]. Recent studies have proposed self-regulation as an explanatory cross-diagnostic variable, where different diagnostic categories converge [13,14,15,16].

Research has evolved, however, allowing us to move from Brown’s initial general self-regulation concept [17] toward new conceptions that imply both (1) a gradation in types of self-regulated behavior, and (2) the consideration of both internal and external factors in one’s self-regulation. A heuristic that takes into account this new conception is included in the theory presented below.

Based on this new model and on previous studies by De la Fuente et al. using Brown’s Self-Regulation Questionnaire, specific scales have been developed to examine these different aspects of regulation [18].

### 1.2. The Theory of Self Regulation vs. External Regulation

The theoretical model of SR versus external regulation (ER) [9] is complementary to Brown’s prior theory [17] of self-regulated behavior. In both theories, self-regulated behavior is a variable that makes possible skill acquisition, that is, it refers to a self-directed process where a subject transforms his or her mental abilities into practical skills in multiple contexts [19].

SR vs. ER theory [9,20,21,22,23,24,25,26], when applied to health, seeks to explain the combination of external and internal conditions that predispose adequate behavior in health situations. SR vs. ER theory emerged in the field of self-regulated learning [27,28], but has evolved toward the study of general self-regulated behavior [4] and toward self-regulated health behavior, of university students in particular [25,29]. It can be described as follows.

#### 1.2.1. Self-Regulation, Non-Regulation and Dys-Regulation of Health Behavior

Self-regulated health behavior is understood as the ability to plan and manage one’s own health-related behavior in a flexible way, in accordance with desired outcomes. The initial theoretical conception of self-regulation, non-regulation and dys-regulation (SR-NR-DR) has been applied to three levels of health regulation. Individuals’ level of competence in the area of health behavior can be classified into three regulatory levels. Levels 1, 2, 3 indicate a descending gradation (high-medium-low) in level of personal health regulation: self-regulated health (SRH), the most proactive self-regulatory level; non-regulation of health (NRH), a middle level that is not proactive; and dys-regulation of health, where negative proactive behaviors may even appear (DRH) [8,21,24,30]:


(1)*Self-Regulated Health behavior* (SRH) reflects an individual’s positive proactivity, that is, active and adequate management of their own health-related behavior [28,31]. Previous research has shown that people can have different degrees of self-regulation [8,20,32].(2)*Non-Regulation of Health behavior* (NRH) refers to lack of proactivity and the absence of personal health-regulating behaviors. This is the conceptual equivalent of reactivity [19,25,33]. In this case, the student is fully dependent on whatever external regulation is provided by the context.(3)*Dys-Regulated Health behavior* (DRH) refers to a degree of negative proactivity in one’s behavior, that is, active but inadequate behavior. The individual may find advantages in such dysregulation, which enables them to avoid the effort involved in positive, proactive health regulation. Examples include self-handicapping strategies, procrastination in matters of health or other risky behaviors [34,35,36,37,38,39].


Based on Zimmerman’s model [27,28], Principle 1 of SR vs. ER theory [9,20,40] suggests a graded continuum of health-behavior regulation. In this framework, specific behaviors carried out before, during and after a given health-related behavior have been described. See Table 1.

#### 1.2.2. Contexts of External Regulation, External Non-Regulation, and External Dys-Regulation of Health Behavior

A context that is externally regulatory in health is defined as providing environmental stimuli that predispose, to a greater or lesser degree, self-regulated health behavior. Similar to internal, self-regulation, external contexts can also be classified into three levels of regulation of health behavior. In external regulation of health behavior, the context encourages the individual to make adequate use of the different health-regulating competencies [10,22,23,24,25,26,41]: externally-regulated health behavior (ERH) is the most proactive level of a health-promoting context; external non-regulation of health behavior (ENH) is a context that does not offer external support to the student; finally, a context which is externally dysregulatory toward health behavior (EDH) is characterized by unhealthful factors (stress triggers in the teaching process, prompting negative achievement emotions, etc.):


(1)*Externally-regulated health behavior (ERH).* In this context, individuals are encouraged to show positive or adequate proactivity and to practice self-regulation of health [42]. External indicators or promptings promote and increase the likelihood of self-regulated health behaviors. These may take the form of antecedents (patterns, norms, limits, expectations of successful health regulation, value given to health regulation) or of contextual consequences (positive and negative contingencies that favor health regulation, adaptation, etc.). Positive events that result in regulated health are strongly predicted in this context [43]).(2)*External non-regulation of health behavior (ENH).* The context does nothing to encourage one’s personal health regulation, but it does not contain health-dysregulating factors. No external indicators or promptings encourage self-regulated or dysregulated health behaviors, nor do they increase the likelihood of one or the other. A non-regulatory context requires the individual to employ a moderate level of their own health-regulating behavior, given that contextual elements offer no direction. Positive events that result in regulated health are not predicted in this context [44]).(3)*External dys-regulation of health (EDH),* actively promoting non-adaptive health behaviors. Inadequate or negative proactivity is encouraged in this type of context. There are many external indicators that encourage and increase the likelihood of active, dysregulating behavior. This behavior is also stimulated by contextual antecedents (modeling, rules, limits, negative expectations of self-regulation, negative value given to self-regulation, etc.) and by contextual consequences (positive and negative contingencies, molding, etc.) that encourage dysregulation. A great deal of effort is required from the individual who wishes to practice health-regulating behavior in this context. This context is highly predictive of negative events that result in dysregulated health.


In this case, Principle 2 of SR vs. ER Theory [9,20,40], also based on Zimmerman’s model [26,27], establishes the specific external, contextual characteristics that encourage or hinder self-regulation. The behavioral sequence is the same as in Zimmerman’s model: before-during-after. See Table 2.

#### 1.2.3. Combinations of Self/External Regulation, Non-Regulation, and Dys-Regulation of Health

Principle 3 of SR vs. ER [9,20] establishes interactivity between the previous levels. Prior research presented a heuristic of regulation combinations that can explain variability in the academic sphere and is able to explain and predict different academic learning behaviors [9,22,23,24,25,26]. In similar fashion, the student’s level of self-regulated health behavior combines with levels of external regulation offered by the context. It is plausible to assume that behavioral contexts may interact with personal levels of self-regulation in order to explain diversity in student health behaviors. An illustration from the coronavirus disease 2019 (COVID-19) situation was recently published (de la Fuente, Kauffman, Dempsey and Kauffman, 2021) [40].

### 1.3. Psychological Reactance

Psychological reactance is a motivational state that individuals may adopt when they perceive that a freedom is eliminated or threatened. It is an unpleasant provocation that motivates them to act to restore their autonomy [45,46,47]. Laird and Frazer [5] found that high levels of behavior problems can predict high levels of reactance and perceived psychological control. Likewise, high levels of reactance can lead adolescents and youths to interpret positive or ambiguous parenting behaviors as compatible with psychological control. Hence, when individuals experience psychological reactance, this may indicate a perception that the threatened behaviors or thoughts are more attractive, producing a desire to reestablish them [46,48].

Empirically, psychological reactance has been negatively correlated to self-assessments of happiness and indices of self-esteem, and positively correlated to perceived loneliness and fear of failure. Moreover, persons with psychological reactance tend to have more violent relationships [49,50]. Hong and Faeedda [51] indicated that a high level of psychological reactance could explain a person’s motives for resisting social influence, the invasion of their personal space, or influence from psychotherapy. On the other hand, Steindl and collaborators [45] indicated that psychological reactance can be triggered vicariously through indirect reactions; for this reason, reactance is also studied from a contextual and cultural standpoint. Reactance has been investigated from the spheres of neurophysiology, social psychology, clinical psychology, the psychology of motivation, and so on. Psychological reactance shows obvious potential as a mediator in human interaction, clearly warranting more in-depth study as a critically important construct for both scientists and professionals.

### 1.4. Student Health

Health is a state of social, mental and physical well-being, not merely the absence of disease. It is important to investigate student health, given that half of all mental disorders start by the age of 14; in most cases, however, they are neither detected nor treated [3,52,53]. Physical health ranges across a spectrum from high levels of energy with no chronic conditions or symptoms, to the opposite end of having severe disability. Some of the practices relating to the physical health of youth people are: regularity in eating, regularity of physical activity, and the use of psychoactive substances like alcohol and tobacco [42,54].

Psychologically, we speak of health when the student enjoys a set of characteristics such as positive emotions, enthusiasm, vitality, self-determination, and subjective perception of well-being [55,56]. On the other hand, a significant quantity of research indicates the importance of university students’ self-assessed health; when this assessment is unsatisfactory, it is associated with sedentarism, headaches and insomnia and with lower quality of life [57,58,59]. In the student population, a low level of well-being has been related to low academic achievement and high levels of procrastination [4,60]. Recent research has studied how physical and psychological health mutually affect one another [61], so student health can be approached from the standpoint of well-being, as proposed by the WHO. In relation to the aforementioned variables, a positive correlation has been found between levels of self-regulation and one’s degree of physical and psychological well-being [4].

### 1.5. Objectives and Hypotheses

Given the prior evidence, the general aim of the current study was to establish possible linear associations and structural prediction relationships between self-regulated health behavior and external regulation of health behavior, with respect to psychological reactance and student health.

The association hypotheses were as follows:

**Hypothesis 1.** 
*Self-regulated behavior (SR) is expected to show a positive association with self-regulation of health (SRH), and a negative association with non-regulation (NRH) and dys-regulation of health (DRH); this directionality will be identical in the case of the external regulatory factors, that is, external regulation of health (ERH), external non-regulation of health (ENH) and external dysregulation of health (EDH).*


**Hypothesis 2.** 
*In a complementary way, self-regulation of health (SRH) is expected to have a negative association with non-regulation of health (NRH) and dys-regulation of health (DRH); also, external regulation of health (ERH) will show a negative association with external non-regulation of health (ENH) and external dys-regulation of health (EDH).*


**Hypothesis 3.** 
*External non-regulation of health (ENH) will be positively related to personal non-regulation of health (NRH); and external dys-regulation of health (EDH) will be positively associated with personal dys-regulation of health (DRH).*


**Hypothesis 4.** 
*External regulation of health (ERH) will be positively associated with student health (SH); dys-regulation of health (DRH) will be positively associated with psychological reactance (REACT) and negatively with student health (SH); and external regulation of health (ERH) will be negatively associated with reactance (REACT) and positively with student health (SH).*


**Hypothesis 5.** 
*Psychological reactance (PR) will be negatively associated with student health (SH) and with self-regulation of health (SRH).*


The prediction hypotheses were as follows:

**Hypothesis 6.** 
*Each level of external regulation will predict the same level in self-regulation: external regulation of health (ERH) will predict internal, self-regulation of health (SRH); external non-regulation of health (ENH) will predict internal non-regulation of health (NRH); external dysregulation of health (DRH) will predict internal dysregulation of health (DRH).*


**Hypothesis 7.** 
*External contexts that are non-regulating or dysregulating toward health, together with personal dysregulation of health behavior, will jointly predict psychological reactance, and the latter will negatively predict general self-regulation and student health (physical and psychological); by contrast, external health-regulating contexts in conjunction with personal self-regulated health will positively predict general self-regulation and also student health.*


## 2. Method

### 2.1. Participants

Participating in the study were 269 undergraduate students from different universities. The percentage of responses from the students invited to participate was 98.5% (*n* = 265). The sample contained students enrolled in psychology, primary education, and law degrees; 66.9% were women and 18.3% were men. Average age was 22.82 years (sd = 2.29), age range was from 18 to 29. The sample could not be randomized, hence sampling was incidental and non-probabilistic. The students completed self-reports in reference to their health behaviors. All participation was anonymous and on a voluntary basis.

### 2.2. Instruments

*Self-Regulation Behavior.* Measured by the Short Self-Regulation Questionnaire (SSRQ) [62]. The Spanish adaptation had been previously validated in Spanish samples, and showed acceptable validity and reliability, with values similar to the English version. The Short SRQ comprises [18,63] four factors: goal setting-planning (I usually keep track of my progress toward my study goals), perseverance (I am able to resist temptation), decision making (I have so many plans that it’s hard for me to focus on any one of them) and learning from mistakes (I usually only have to make a mistake once in order to learn from it). There are 17 items (all of them with saturations >0.40), and a consistent confirmatory factor structure in this sample—Chi-square = 439.699, df = 113, *p* < 0.001; Ch/df = 3.89; standardized root mean square residual (SRMR) = 0.088; comparative fit index (CFI) = 0.959, GFI = 0.951, AGFI = 0.97, root mean square error of approximation (RMSEA) = 0.076. Internal consistency was acceptable for the questionnaire total (α = 0.87) and for the factors: goal setting-planning (α = 0.79), decision making (α = 0.72), perseverance (a = 0.75), and learning from mistakes (α = 0.72).

*Self-Regulation vs. External Regulation of Health.* This questionnaire [64] has a structure of 6 subscales with six items each. It assesses health-regulating aspects of both the person and the context. The behaviors assessed may be regulatory [internal: I think consciously about my health needs, external: The social context that I live in (family, environment, friends) helps me plan my health-related behavior by setting goals and objectives]; non-regulatory [internal: It is not necessary to make decisions in order to achieve changes in my health-related behaviors, external: The social context that I live in (family, environment, friends) gives me the idea that you do not need to make specific decisions in order to achieve changes in your health-related behaviors]; or dysregulatory (internal: It does not make sense to change your health-related behavior, if that takes away from your enjoyment and satisfaction, external: The social context that I live in (family, environment, friends) helps me enjoy myself to the fullest, since it does not press me to change my health-related behavior, but rather to do what I feel like, if that makes me happy and live fully]. Factor structure, as analyzed in this sample, is consistent [Chi Square = 1348.005, df = 583, *p* < 0.001; Ch/df = 2.379; RMSR = 0.035; NFI = 0.967; RFI = 0.954; incremental fit index (IFI) = 0.902; TLI = 0.967; CFI = 0.978; RMSEA = 0.70]. Total reliability values were also acceptable [alpha total = 0.776]. Subscale consistency was also acceptable: SRH = 0.847; NRH = 0.779; DRH = 0.769; ERH = 0.900; ENH = 0.761; EDH = 0.828. See Table 3 for a conceptual listing of the types of subscales and their hypothesized relationships.

*Psychological reactance* measured by the *Escala de Reactancia Psicológica* (Scale of psychological reactance) [40]. The scale structure consists of four subscales; however, our exploratory and confirmatory analysis showed three scales in our sample. Construct structural values were adequate in this sample [Chi-square = 100.301, df = 41, *p* < 0.001; Ch/df = 2.446; RMR = 0.042; CFI = 0.929 GFI = 0.917; IFI = 0.918; TLI = 0.927; CFI = 0.915; RMSEA = 0.073]. Total reliability values were also acceptable [alpha total = 0.734]. Some examples of scale items are: Advice and recommendations usually prompt me to do just the opposite, and, Regulations trigger a sense of resistance in me.

*Student Health* [4]. Measured by the *Cuestionario de Salud Académica* (Student Health Inventory). This self-report scale assesses students’ physical health (I have good eating habits/I sleep well) as well as psychological health (I feel depressed by studies, I properly balance homework with recreational activities). The structure consists of two subscales. Construct structural values were adequate in this sample [Chi-square = 41.385, df = 19; *p* < 0.001; Ch/df = 2.446; RMR = 0.021; CFI = 0.946; GFI = 0.947; IFI = 0.941; TLI = 0.911; CFI = 0.940; RMSEA= 0.073]. Total reliability values were also acceptable [alpha total = 0.751].

### 2.3. Procedure

Students were invited to participate in the study voluntarily. After reading the informed consent statement, students completed the scales using an online platform that ensured response anonymity. Questionnaires were completed online. The R&D Project was approved by the Ethics Committee of the University of Navarra (ref. 2018.170). Compliance with the deontological norms of psychology was assured.

### 2.4. Data Analyses

Using an ex post facto, transversal design [65], three types of analyses were conducted. The usual assumptions of regression analysis were tested beforehand. First, we explored the quality of the data by examining the existence of outliers and missing cases. The method used for detecting univariate outliers consisted of calculating the typical scores of each variable and considering cases with Z scores outside the ±3 range to be potentially atypical cases [65]. On the other hand, the Mahalanobis distance (D^2^) was used to detect atypical combinations of variables (atypical multivariate cases), a statistical measure of the multidimensional distance of an individual, with respect to the centroid or mean of the observations given [66]. This procedure detects significant distances from the typical combinations or centroids of a set of variables. The literature suggests removing univariate and multivariate outliers, or recodifying them to the nearest extreme score [67]. The procedure was carried out using SPSS (26, IBM, Armonk, NY, USA) which provides a specific routine for missing values analysis that determines the magnitude of missing values and whether they are presented in a systematic or random manner.

Assumptions related to sample size, independence of errors, univariate and multivariate normality, linearity, multicollinearity, recursion, and interval measurement level were also evaluated, presenting acceptable levels of reliability. Regarding the sample size, inclusion of 10–20 cases per parameter is recommended, and at least 200 observations [68]. Independence of errors refers to the fact that the error term of each endogenous variable must not be correlated with other variables. In order to determine univariate normality, we examined the distribution of each observed variable, and its indices of asymmetry and kurtosis. Asymmetry values greater than 3 and kurtosis greater than 10 suggest that the data should be transformed [68]. On the other hand, values less than 70 on the Mardia multivariate index indicate that distance from the multivariate normality is not a critical deterrent to this analysis [69]. Although one of the assumptions is the level of interval measurement, in some cases, variables measured at a nominal or ordinal level are used, as long as the distribution of scores, particularly of the dependent variables, is not markedly asymmetric [67].

The assumption of multicollinearity was analyzed by examining bivariate correlations between variables, since a correlation of 0.85 or higher would indicate difficulty in fulfilling this assumption. The model should be recursive, so causal influences must be one-directional and without retroactive effects. Finally, it is recommended that the measuring instruments show at least moderate reliability properties. This aspect was also fulfilled (see instruments section).

A power value of 0.80 was determined as acceptable. The power of a statistical test is related to: (1) sample size *n*; (2) level of significance alpha: 5% was assumed, that is, a 95% confidence level (1-alpha); (3) effect size *d* or *r*: these measures indicate the relationship between variables (correlation coefficient). Low power might indicate a small sample size, a smaller alpha, or a small effect size, while the opposite can be true of high power.

As a preliminary analysis, we checked for normal sample distribution using the Kolmogorov–Smirnoff test for dependent variables. We also used the Hoelter Index to determine sample size adequacy [65]. In addition, we conducted analyses of linearity and atypical values, missing and influential cases, as well as critical values of multivariate normality; recommended values for the multivariate index of kurtosis, or Mardia coefficient, are less than 70 [70].

For Hypothesis 1, Pearson bivariate correlations were carried out. For Hypothesis 2, we used multiple regression analysis. For Hypothesis 3, we used predictive analyses of structural equations, or SEM models. For this purpose, we followed the recommendations of Hu and Bentler [71], where a model shows adequate fit to the observed data when the ratio of the chi-square statistic to its degrees of freedom is less than five, the values of RMSEA and SRMR are <0.08, and NNFI (non-normed fit index), IFI and CFI are >0.95 [72]. If sample sizes are equal to or less than 250 participants, Hu and Bentler [71] suggest using only the CFI and SRMR fit indices (this was not our case). As an estimation method, the robust maximum likelihood method was used, which allows the use of polychoric correlations; their use is more suitable in variables with the foregoing characteristics of high normality indices and multivariate kurtosis, and a clearly ordinal nature [73]. Reliability of the model dimensions, of the total, and of each of the proposed factor structures was also examined by calculating Cronbach’s alpha. The software programs used to perform these analyses were SPSS 26 [74] for the reliability analyses, and AMOS v. 23 [75] for the confirmatory factor analyses and SEM model.

## 3. Results

### 3.1. Previous Analyses

The results were adequate for verifying assumptions of normality in the different indicators analyzed. See Table 4:

The Kolmogorov–Smirnov (*p* < 0.200) and Shapiro-Wilk (*p* < 0.208) tests were nonsignificant in all cases. Mardia coefficient or multivariate kurtosis was appropriate (<0.70).

### 3.2. Linear Association

Correlations between general SR and types of health regulation behavior (internal and external).

A significant, positive correlation appeared between SR and internal (SRH) and external regulation (ERH) of health (*r* = 0.405, *p* < 0.001; *r* = 0.365, *p* < 0.001 respectively). By contrast, SR showed a significant negative correlation with personal NRH (*r* = −0. 321, *p* < 0.001) and ENH (*r* = −0. 341, *p* < 0.001), and with personal DRH (*r* = −0.140, *p* < 0.05) and EDH (*r* = −0.231, *p* < 0.001. See the first column of Table 5.

In complementary fashion, there was a similar trend with SRH, since it showed a significant, negative correlation with NRH (*r* = −0.418, *p* < 0.001). ERH, likewise, showed a significant, negative correlation with ENH and dysregulation of health (EDH) (*r* = −0.336, *p* < 0.001; *r* = −0. 211, *p* < 0.001 respectively). However, the negative correlation between SRH and DRH was not significant.

Other results of interest refer to significant, positive correlations between SRH and ERH (*r* = 0. 310, *p* < 0.001), and significant, negative correlations between SRH and internal (NRH) and external non-regulated health (ENH) *(r* = −0.418, *p* < 0.001; *r* = −0.139, *p* < 0.05, respectively). Similarly, there were positive correlations between internal and external non-regulation of health (*r* = 0.395, *p* < 0.001), and between internal and external dys-regulation of health (*r* = 0.492, *p* < 0.001). In other words, a significant, positive correlation was found between levels of internal and external regulation, that is: SRH and ERH, NRH and ENH, DRH and EDH. See Table 5.

Regarding psychological reactance, it correlated positively with DRH (*r* = 0.341, *p* < 0.001) and negatively with ERH (*r* = 0.232, *p* < 0.001) and with student health (SH) (*r* = −0.151, *p* < 0.05). By contrast, student health (SH) showed a significant, positive association with internal and external regulation of health (SRH and ERH) (*r* = 0. 183, *p* < 0.001; *r* = 0.252, *p* < 0.001, respectively) and a negative association with external non-regulation and dysregulation of health (ENH and EDH) (*r* = −0.201, *p* < 0.001; *r* = −0.156, *p* < 0.001, in that order) and with internal non-regulation (NRH) (*r* = −0.123, *p* < 0.05). It is noteworthy that student health (SH) showed a significant, positive association with self- and externally-regulated health behavior (SRH, ERH) and a negative association with internal and external non-regulation and dysregulation of health (NRH, ENH, DRH, EDH). See Table 6.

### 3.3. Linear and Structural Prediction

#### 3.3.1. Structural Models

Three structural models were tested. Model 1 evaluated the predictive effect of gender on ERH with respect to SRH, reactance and student health (SH). Model 2 evaluated the predictive effect of ERH with respect to SRH, and the predictive value of these two with respect to psychological reactance (REACT) and student health (SH). Model 3 tested the predictive value of the 6 factors of internal and external health regulation with respect to general self-regulated behavior (SR), reactance (REACT), and student health (SH). The third model showed the best statistical standards. See Table 7.

#### 3.3.2. Direct and Indirect Effects

Regarding direct predictive effects from the self- and context-regulation factors, results showed that F1 (SRH) negatively predicted F2 (NRH) (−0.550), which in turn was a predictor of F3 (DRH) (0.203). In complementary manner, F4 (ERH) positively predicted F1 (SRH) (0.316) and negatively predicted F5 (ENH) (−0.409), while F5 (ENH) positively predicted F2 (NRH) (0.433), and F6 (EDH) positively predicted F3 (DRH) (0.584).

Factors F1 (SRH) and F4 (ERH) predicted SR (0.440; 0.226, respectively). Psychological reactance was positively predicted by F3 (EDH) (0.543) and negatively by F4 (ERH) (−0.208). General self-regulation of behavior (SR) strongly predicted student health (SH) (0.790), while psychological reactance (REACT) negatively predicted student health (SH) (−0.322). See Table 8 and Figure 1.

Indirect predictive effects are particularly interesting because they further explain the results above (see Table 9). F1 (SRH) and F4 (ERH) negatively predicted F3 (DRH) (−0.111; −0.216, correspondingly) and psychological reactance (REACT) (−0.060 and −0.117), while positively predicting self-regulation behavior (SR) (0.019; 0.243) and total student health (SH) (0.363; 0.371, respectively). It is noteworthy that F4 (ERH) had greater predictive strength, and was also a negative predictor of F2 (NRH) (−0.351). In complementary manner, F2 (NRH) and F5 (ENH) negatively predicted self-regulation behavior (SR) (−0.035 and −0.077, in that order) and total health (SH) (−0.028; −0.061, correspondingly), and positively predicted psychological reactance (0.110 and 0.239). In this case again, it is the context factor of non-regulation, F5 (ENH), that has more predictive strength; it also positively predicted F3 (DRH) (0.441). In addition, psychological reactance negatively predicted student health (−0.254).

Regarding the indirect predictive effect of the factors on items of other factors, we point to the negative effect of F1 (SRH) on items 7–12 of F2 (NRH) (−0.313 and −0.259, respectively) and on items 13–18 of F3 (DRH) (−0.066 and −0.063). In addition to having these same effects, F4 (ERH) also had a positive effect on items from F1 (SRH). F2 (NRH) had similar effects on F5 (ENH). F2 (NRH) had a positive effect on items 13–18 of F3 (DRH) (0.119; 0.115, in that order), while F5 had additional positive effects on items 7–12 of F2 (NRH) (0.246 and 0.204). Besides the two preceding factors, F3 (DRH) and F6 (EDH) also showed negative predictive effects on factors of general self-regulation (SR) and of health, while their effects were positive for psychological reactance. Finally, self-regulation behavior (SR) indirectly and positively predicted factors of health, while reactance negatively predicted factors of self-regulation (SR) and student health (SH).

## 4. Discussion

These results allowed us to partially validate the relationships between constructs that are proposed in SRL vs. ERL Theory [9], using specific instruments for this purpose. Some of these relationships between constructs had already been shown using prior instruments. Evidence has shown that the combination of levels of students’ self-regulation and of regulatory teaching account for variability in achievement emotions [22,25], coping with stress [24], learning approaches [26], and achievement [21,76]. Notwithstanding this, it was necessary to demonstrate these relationships with specific measuring instruments in order to corroborate the prior evidence. The new instruments assess degree of self-regulation, non-regulation and dysregulation (internal and external), proposed as specific theoretical constructions pertaining to the aforementioned theory. This study has generated empirical evidence about the convergent validity of the questionnaire on Self-Regulation vs. External Regulation of Health. To this end, we have examined its relations with measures of Self-Regulation behavior, Psychological reactance and Student health.

### 4.1. Discussion of Hypotheses

Regarding the association hypotheses, the evidence shows that general self-regulation behavior (SR) is positively associated with SRH, as well as negatively associated with NRH and with DRH, whether we speak of personal or contextual regulation (Hypothesis 1). This constitutes an external validation of these new constructions that expand on the prior, classic construct of SR, amply validated in the literature [4,63]. In complementary manner, SRH appeared with the same tendency, that is, negatively associated with personal NRH and with DRH behaviors. In the same way, External Regulation of Health was negatively associated with ENH and EDH.

This result is of particular interest for two reasons: (1) on the one hand, it allows us to theoretically acknowledge the category of dysregulatory behavior, which has not been sufficiently addressed in research on health regulation behaviors, even though it is a concept used in the field of mental health [77,78]; (2) on the other hand, it allows us to theorize the behavioral sequence between self-regulation of health and dysregulation of health. It seems reasonable to assume that a certain self-regulating health behavior may evolve toward a state of non-regulation, when an individual stops making the effort to self-regulate—studies on ego depletion are an example [79,80]. In addition, it seems to suggest that non-regulation is a previous step that is positively associated with dysregulation.

Other interesting associations had to do with the significant negative relationship between psychological reactance and student health. Although reactance had already been associated negatively with self-assessments of happiness and self-esteem, and positively with perceived loneliness and fear of failure [49,50], the present study used a broader model of health, measuring both psychological and physical constructs. Personal dysregulation of health was positively correlated to psychological reactance, while external regulation of health was negatively related. This result is similar to that reported by Laird and Frazer [5], where psychological reactance was related to behavioral problems; the present research adds the complementary aspect of relating it to problems in student health.

For its part, student health had several significant relationships, being positively associated with internal and external health regulation, and negatively with internal and external non-regulation and dysregulation of health. These results underscore not only the importance of students regulating their own behavior in order to attain optimal states of health, but also that one’s state of health is related to how well the context (family, friends, community) encourages self-regulating choices in health behavior.

Regarding linear prediction hypotheses using the structural model, one especially interesting result was the prediction from context to health behavior (SRH, NRH, and DRH) in the three types. Certain prior studies have already documented the importance of context in health behaviors [81,82]. Also, of interest was the negative prediction of self-regulating health behavior on non-regulating health behavior, and the positive prediction of non-regulating health behavior on dysregulating health behavior. This predictive sequence of health behavior could become a powerful explanatory heuristic that addresses both personal and contextual regulation of healthful conduct. Prior evidence has shown a sequential evolution—not always made explicit—between lowering one’s level of regulation, to not regulating oneself, to dysregulation, where these must be considered differentially in the analysis of problematic health behaviors [83]. Contexts can also follow this sequence, from weakening their regulatory aspect, to becoming non-regulatory, and finally, encouraging dysregulation of health. The highest levels in both sequences predict the highest level of student health, while the second and third levels predict successively lower levels of student health. These results, though novel, corroborate certain prior evidence referring to the role of self-regulation in one’s personal health behavior [4,32,84,85,86].

The innovative aspect of the present proposition is that it more precisely defines types of regulatory behaviors as a heuristic for predicting student health: SRH-NRH-DRH. This conceptual and empirical proposition also includes context in the progressive regulatory categorization: ERH-ENH-EDH. Previous evidence has recognized the concept of health dysregulation, alluding to both physical factors [87] and behavioral factors [83]. However, the concept of non-regulation, as an intermediate link between self-regulated and dysregulated health, has not yet been the object of analysis.

Moreover, there has been little analysis of the role of context in promoting or not promoting health behaviors. While context has usually been assumed to play a meaningful role in health, no typology of this role was defined. The present study classifies it along this continuum: ERH-ENH-EDH. Plentiful evidence has suggested the relevance of context in promoting health, though without this analytical breakdown [85,88].

### 4.2. Limitations

The present study also has limitations to be analyzed. On one hand, the sample is limited; the number of students is insufficient for making high-level inferences to the population, even though a large number of countries are represented. On the other hand, the initial validation of the instruments was conducted using this same sample. For this reason, despite the good statistical fit, the instruments measuring self-regulation of health (SRH-NRH-DRH) and external regulation of health (ERH-ENH-EDH) should continue to be studied and revalidated. Consequently, the results should be taken with caution.

### 4.3. Goodness and Future Research

This study contributes three highly relevant types of evidence. (1) There is preliminary empirical evidence regarding the hypothesized gradation of types of regulatory behavior and context in the SR vs. ER theoretical model [9], in this case applied to health (regulatory, non-regulatory and dys-regulatory), and positioned alongside the classic construct of self-regulation behavior (SR). We have documented that this gradation is measurable and has empirical consistency. (2) There is evidence of the individual and combined predictive value of the regulatory characteristics of the context and of the subject for predicting students’ psychological reactance and health behaviors. As predicted by the theory and prior evidence, context has an essential effect in inducing each regulation type in subjects. In other words, regulatory contexts predict self-regulated health behavior, non-regulatory contexts produce nonregulated health behavior and dysregulatory contexts predict dysregulatory behavior. (3) Finally, there is evidence that psychological reactance is jointly predicted by a dysregulatory context that induces dysregulation of health behavior, as well as by the subject’s own level of behavioral dysregulation. Psychological reactance should, therefore, be considered a behavioral correlate of dys-regulation, a non-protective variable in university students’ health behaviors. These results as a whole lend empirical support to recent studies that propose self-regulation as an explanatory cross-diagnostic variable, where different diagnostic categories converge [13,14,15,16], taking the combined explanatory view of *subject x context*.

Future research should validate the factorial invariance of these relationships in other contexts, such as personality variables, education, and other health problems. On the one hand, the adequacy of this regulatory categorization (SRH-NRH-DRH; ERH-ENH-EDH) should be verified for its utility in characterizing different behavior problems. On the other hand, cross-cultural studies should be conducted to provide evidence for the intercultural validity of this categorization. These preliminary results once again lend empirical support to the SR vs. ER Theory [9].

### 4.4. Professional Implications

There are various implications for intervention in university students’ health and self-regulation. First, it is important to identify the types of health regulation in each case, both internal to the student (SRH-NRH-DRH), and external, from the context (ERH-ENH-EDH). It is essential to take into consideration both sides of this reality in order to assess and intervene in student health programs, using an interactive approach. For too long, student health has been considered almost exclusively determined by the student’s own behaviors, to the detriment of contextual considerations [25]. The time has come to begin designing interventions that also address contextual factors, so that these may likewise promote the health of students [88].

## Figures and Tables

**Figure 1 ijerph-18-03793-f001:**
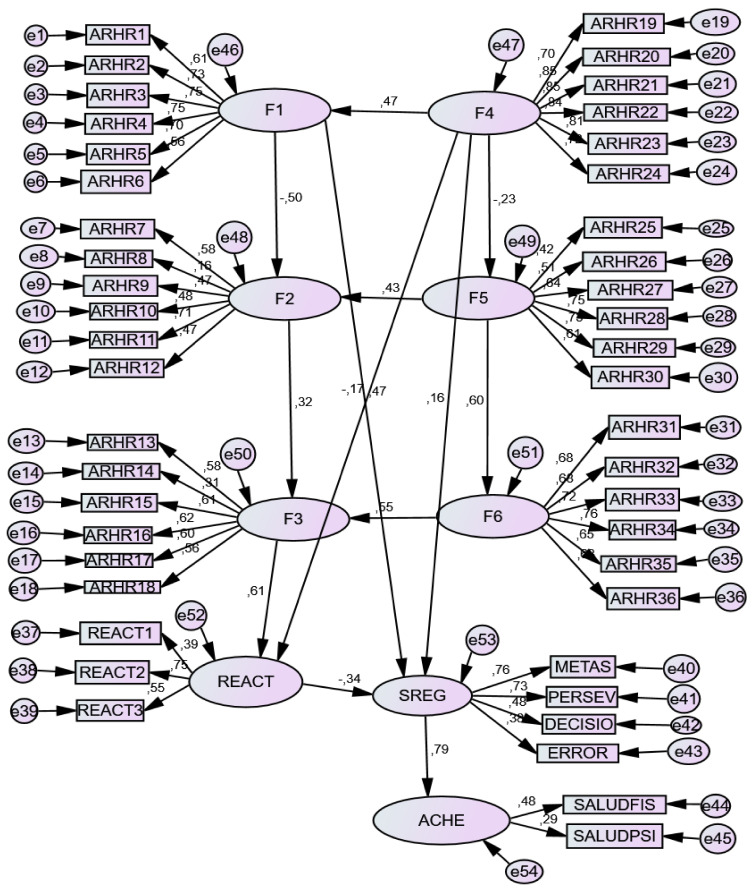
Structural Equations Model of prediction. F1 = Self-Regulation of Health; F2 = Self Non-Regulation of Health; F3 = Self Dys-Regulation of Health; F4 = External Regulation of Health; F5 = External Nonregulation of Health; F6 = External Dysregulation of Health; SR = Self-regulation of general behavior; REACT = Reactance; ACHE = Student Health; ARHR = items of SRH vs. ERH questionnaire; METAS = Goals; PERSEV = Perseverance; DECISIO = Decisions; ERROR = Learning from mistakes; REACT = Factors of reactivity: SALUDFIS = Physical health; SALUDPSI = Psychological health.

**Table 1 ijerph-18-03793-t001:** Conceptual continuum and typologies of personal health regulation.

Characteristics of Health Behavior Regulation	(1) Self-Regulated Health (SRH) POSITIVE PRO-ACTIVITY (+1)	(2) Non-Regulated Health (NRH) RE-ACTIVITY (0)	(3) Dys-Regulated Health (DRH) NEGATIVE PRO-ACTIVITY (−1)
	*Before (planning)* Self-analysis Self-defined goals Self-motivation	*Before (no planning)* No self-analysis No goals No motivation	*Before (planning amiss)* Erroneous self-analysis Erroneous goals Self-demotivation
	*During (self-monitoring)* Self-observation Self-analysis Self-correction	*During (no monitoring)* No self-observation No self-analysis No self-correction	*During (misguided monitoring)* Self-distraction Cognitive self-avoidance Self-handicapping, procrastination
	*After (self-assessment)* Personal reflection Self-attributions Positive self-affect	*After (no assessment)* No reflection No attributions Indifferent in affect	*After (misguided assessment)* Erroneous self-assessment Erroneous self-attributions Negative self-affect
*Type of Activity*	*Self-Regulated Health (SRH)*	*Non-Regulated Health (NRH)*	*Dys-Regulated Health (DRH)*
Health Behavior	SR Health	No norms/limits	Self-induced excesses/deficits

**Table 2 ijerph-18-03793-t002:** Conceptual continuum of external health regulation.

Characteristics of the External Regulatory Context	1. External Regulation of Health (ERH) POSITIVE PROACTIVITY (+1)	2. External Non-Regulation of Health (ENH) NEUTRAL (0)	3. External Dys-Regulation of Health (EDH) NEGATIVE PRO-ACTIVITY (−1)
	*Before (planning)* Introduces tasks Favors well-adjusted goals Favors self-motivation	*Before (no planning)* Does not present tasks Does not suggest goals Does not induce motivation	*Before (planning amiss)* Erroneous tasks Favors self-handicapping goals Induces demotivation
	*During (favors monitoring)* Promotes self-observation Promotes self-analysis Promotes self-correction	*During (indifferent)* No self-observation No self-analysis No self-correction	*During (discourages monitoring)* Promotes self-distraction, Cognitive self-avoidance, Self-handicapping, Procrastination
	*After (prompts assessment)* Promotes self-reflection Promotes adjusted self-attributions Promotes positive adjusted self-affect	*After (ignores assessment)* No reflection No attributions Indifferent in affect	*After (misguides assessment)* Promotes erroneous self-assessment Erroneous self-attributions Promotes maladjusted self-affect
***Example:*** ***alcohol use*** **Context** **(environmental characteristics)**	Externally regulatory Norms/limits on consumption	Non-regulatory No norms/consequences	Dys-regulatory Encourages alcohol abuse

**Table 3 ijerph-18-03793-t003:** Types of self and external regulation of health.

External Regulation	Self-Regulation		
	SRH (+1)	NRH (0)	DRH (−1)
ERH (+1)	*+*	*−*	*−*
ENH (0)	*−*	*+*	*−*
EDH (−1)	*−*	*+*	*+*

**Table 4 ijerph-18-03793-t004:** Preliminary Analyses.

Statistical Index	SR	SRH	NRH	DRH	ERH	ENH	EDH	REACT	SH
Mean	3.437	3.576	2.670	2.564	3.497	2.480	2.254	2.657	3.571
Mean Standard Error	0.035	0.045	0.039	0.043	0.055	0.046	0.048	0.033	0.038
Mode	3.24	3.33	2.83	2.50	3.67	2.50	2.00	2.44	3.60
Standard Deviation	0.588	0.753	0.652	0.716	0.905	0.753	0.800	0.552	0.630
Asymmetry	0.088	−0.168	0.182	0.091	−0.359	0.318	0.443	0.257	−0.262
Standard asymmetry error	0.149	0.149	0.149	0.149	0.149	0.149	0.149	0.149	0.149
Kurtosis	−0.480	−0.143	−0.062	−0.345	−0.157	−0.180	−0.214	0.263	−0.079
Standard kurtosis error	0.296	0.296	0.297	0.297	0.296	0.297	0.296	0.296	0.296
Range	3.09	3.67	3.50	3.67	4.00	3.83	3.67	3.25	3.30
Minimum	1.86	1.33	1.17	1.00	1.00	1.00	1.00	1.00	1.70
Maximum	4.95	5.00	4.67	4.67	5.00	4.83	4.67	4.25	5.00

Note. SR = Self-Regulation behavior; SRH = Self-Regulation of Health behavior; NRH = Non-Regulation of Health behavior; DRH = Dys-regulation of Health behavior; ERH = External Regulation of Health behavior; ENH = External Nonregulation of Health behavior; EDH = External Dysregulation of Health behavior; REACT = Psychological Reactance, total score; SH = Student Health.

**Table 5 ijerph-18-03793-t005:** Bivariate correlations between self-regulation and types of health regulation (internal and external) (*n* = 269).

Variables	SR	SRH	NRH	DRH	ERH	ENH	EDH
SR							
SRH	405 ***						
NRH	−0.321 ***	−0.418 ***					
DRH	−0.140 *	−0.018	0.321 ***				
ERH	365 ***	310 ***	−0.016	−0.097			
ENH	−0.341 ***	−0.139 *	0.395 ***	−0.394 ***	−0.336 ***		
EDH	−0.231 ***	0.032	0.245 ***	0.492 **	−0.211 **	0.499 ***	

Note. SR = Self-Regulation behavior; SRH = Self-Regulation of Health behavior; NRH = Non-Regulation of Health behavior; DRH = Dys-regulation of Health behavior; ERH = External Regulation of Health behavior; ENH = External Nonregulation of Health behavior; EDH = External Dysregulation of Health behavior; * *p* < 0.05, ** *p* < 0.01, *** *p* < 0.00.

**Table 6 ijerph-18-03793-t006:** Correlations between types of internal and external regulation, psychological reactance and health (*n* = 269).

Variable	SRH	NRH	DRH	ERH	ENH	EDH	REACT	SH
REACTAN	−0.035	0.106	0.341 ***	−0.232 ***	0.285 ***	0.284 ***		
SH	183 **	−0.123 *	−0.113	0.252 ***	−0.201 **	−0.156 *	−0.151 *	

Note. SRH = Self-Regulation of Health behavior; NRH = Non-Regulation of Health behavior; DRH = Dys-Regulation of Health behavior; ERH = External Regulation of Health behavior; ENH = External Nonregulation of Health behavior; EDH = External Dysregulation of Health behavior; REACT = Psychological Reactance, total score; SH = Student Health; * *p* < 0.05, ** *p* < 0.01, *** *p* < 0.001.

**Table 7 ijerph-18-03793-t007:** Models of structural linear results of the variables.

Model	Chi^2^	DF	CH/df	SRMR	*p<*	NFI	RFI	IFI	TLI	CFI	RMSEA	HOELT 0.05	HOELT 0.01
1	227.217	51	4.455	0.12	0.001	0.798	0.735	0.724	0.876	0.823	0.086	81	92
2	227.217	51	4.455	0.12	0.001	0.810	0.889	0.820	834	0.818	0.084	181	192
3	2028.759	932	2.77	0.05	0.001	0.944	0.904	0.970	0.938	0.965	0.064	133	137

Note. Model 1: Gender → ERH → SRH → Reactance → Health; Model 2: ERH → SRH → Reactance → Health; Model 3: 6 factors of Health Regulation → SR → Reactance → Health; DF = Degrees of freedom; CH/df = Chi Square/Degrees of Freedom; SRMR = Standardized Root-Mean-square Residual; *p*< = probability level; NFI = Normed Fit Index; RFI = Relative Fit Index; IFI = Incremental Fit Index; TLI = Tucker-Lewis Index; CFI = Comparative Fit Index; RMSEA = Root Mean Square Error of Approximation; HOELT = Hoelter index.

**Table 8 ijerph-18-03793-t008:** Standardized direct effects (default model).

Variables	F1 (SRH)	F2 (NRH)	F3 (DRH)	F4 (ERH)	F5 (ENH)	F6 (EDH)	SR	REACT	SH
F1				0.316					
F2	−0.550				0.433				
F3		0.203				0.584			
F4									
F5				−0.409					
F6					0.604				
SR	0.440			0.226				−0.322	
REACT			0.543	−0.208					
SH							0.790		
SRERH1	0.719								
SHERH2	0.771								
SRERH3	0.719								
SRERH4	0.684								
SRERH5	0.692								
SRERH6	0.579								
SRERH7		0.568							
SRERH8		−0.212							
SRERH9		0.457							
SRERH10		0.527							
SRERH11		0.747							
SRERH12		0.471							
SRERH13			0.589						
SRERH14			0.226						
SRERH15			0.481						
SRERH16			0.549						
SRERH17			0.609						
SRERH18			0.566						
SRERH19				0.768					
SRERH20				0.813					
SRERH21				0.846					
SRERH22				0.808					
SRERH23				0.755					
SHERH24				0.663					
SRERH25					0.411				
SRERH26					0.459				
SRERH27					0.551				
SRERH28					0.723				
SRERH29					0.804				
SRERH30					0.659				
SRERH31						0.714			
SRERH32						650			
SRERH33						0.696			
SRERH34						0.731			
SRERH35						0.572			
SRERH36						0.696			
GOALS							0.733		
PERSEV							0.712		
DECIS							0.448		
MISTAKES							0.654		
REACT1								0.834	
REACT2								0.533	
REACT3								0.641	
HEALTH1									0.497
HEALTH2									0.284

Note. F1 = Self-Regulation of Health; F2 = Self Non-Regulation of Health; F3 = Self Dys-Regulation of Health; F4 = External Regulation of Health; F5 = External Nonregulation of Health; F6 = External Dysregulation of Health; SR = Self-regulation of general behavior; REACT = Reactance; SH = Student health; SRER = items of SRH vs. ERH questionnaire; GOALS = Goals; PERSEV = Perseverance; DECIS = Decisions; MISTAKES = Learning from mistakes; REACT = Factors of reactivity: HEALTH1 = physical health; HEALTH2 = psychological health.

**Table 9 ijerph-18-03793-t009:** Standardized indirect effects (default model).

Variables	F1 (SRH)	F2 (NRH)	F3 (DRH)	F4 (ERH)	F5 (ENH)	F6 (EDH)	SR	REACT	SH
F1									
F2				−0.351					
F3	−0.111			−0.216	0.441				
F4									
F5									
F6				0.247					
SR	0.019	−0.035	−0.175	0.243	−0.077	−0.102			
REACT	−0.060	0.110		−0.117	0.239	0.317			
HEALTH	0.363	−0.028	−0.138	0.371	−0.061	−0.081		−0.254	
SRERH1				0.227					
SHERH2				0.243					
SRERH3				0.227					
SHERH4				0.216					
SRERH5				0.218					
SRERH6				0.183					
SRERH7	−0.313			−0.075	0.246				
SRERH8	0.117			−0.160	−0.092				
SRERH9	−0.251			−0.185	0.198				
SRERH10	−0.290			−0.262	0.228				
SRERH11	−0.411			−0.165	0.323				
SRERH12	−0.259			−0.165	0.204				
SRERH13	−0.066	0.119		−0.127	0.260	0.344			
SRERH14	−0.025	0.046		−0.049	0.100	0.132			
SRERH15	−0.054	0.098		−0.104	0.212	0.281			
SRERH16	−0.061	0.111		−0.118	0.242	0.321			
SRERH17	−0.068	0.123		−0.131	0.269	0.356			
SRERH18	−0.063	0.115		−0.122	0.249	0.331			
SRERH19					0.260	0.344			
SRERH20					0.100	0.132			
SRERH21									
SRERH22									
SRERH23									
SHERH24									
SRERH25				−0.168					
SRERH26				−0.188					
SRERH27				−0.225					
SRERH28				−0.296					
SRERH29				−0.329					
SRERH30				−0.270					
SRERH31				−0.176	0.431				
SRERH32				−0.161	0.393				
SRERH33				−0.172	0.420				
SRERH34				−0.181	0.442				
SRERH35				−0.141	0.346				
SRERH36				−0.162	0.396				
GOALS	0.337	−0.026	−0.128	0.344	−0.056	−0.075			
PERSEV	0.327	−0.025	−0.124	0.334	−0.055	−0.073		−0.236	
DECIS	0.206	−0.016	−0.078	0.210	−0.035	−0.046		−0.229	
MISTAKES	0.300	−0.023	−0.114	0.307	−0.050	−0.067		−0.144	
REACT1	−0.050	0.092	0.453	−0.271	0.200	0.265		−0.211	
REACT2	−0.032	0.059	0.289	−0.173	0.127	0.169			
REACT3	−0.039	0.070	0.348	−0.208	0.153	0.203			
HEALTH1	0.180	−0.014	−0.069	0.185	−0.040	−0.040	0.393	−0.127	
HEALTH2	0.103	−0.008	−0.039	0.105	−0.023	−0.023	0.224	−0.072	

Note. F1 = Self-Regulation of Health; F2 = Self Non-Regulation of Health; F3 = Self Dys-Regulation of Health; F4 = External Regulation of Health; F5 = External Nonregulation of Health; F6 = External Dysregulation of Health; SR = Self-regulation behavior; REACT = Reactance; SH = Student Health; SRERH = items of SRH vs. ERH questionnaire; GOALS = Goals; PERSEV = Perseverance; DECIS = Decisions; MISTAKES = Learning from mistakes; REACT = Factors of reactivity: HEALTH1 = Physical health; HEALTH2 = Psychological health.

## Data Availability

Data is available on request from the corresponding author.

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
