# Peer review of "Regulation/Non-Regulation/Dys-Regulation of Health Behavior, Psychological Reactance, and Health of University Undergraduate Students"

_ijerph, 2021, doi:10.3390/ijerph18073793_

Round 1

Reviewer 1 Report

Some comments for consideration:

  1. Please define the term used, e.g., what is self-regulation and what is behavioral self-regulation?
  2. What is the difference between self-regulation and behavioral self-regulation? The authors seem to use them inter-changeably. Please be consistent with the term used.
  3. What is ERL? There is no term expansion in the beginning and no explanation.
  4. Please check the manuscript thoroughly. There is no Table 6.
  5. There is a need to describe the results more, instead of just showing all the result tables.
  6. There is a need to strengthen the discussion section. 
  7. There is no strong contribution highlighted in the study. 
  8. Overall the paper has to be strengthened. It lacks clear directions and significance of findings.
  9. An example of the lack of understanding on the study: "Also of interest was self-regulatory behavior negatively predicting nonregulatory behavior, and the positive prediction of nonregulatory behavior toward dysregulatory"; this point doesn't make much sense and there is no explanation or support given.
  10. Please ensure consistency and coherence throughout the paper.

Author Response

Please, see the attach. Thanks

Reviewer 2 Report

This research examines the possible linear associations and structural prediction relationships between self-regulated health behavior and external regulation of health behavior, with respect to psychological reactance and student health. The results allow to partially validate the relationships between constructs that are proposed in Self-Regulation Learning vs. Externally Regulated Learning Theory.

The major strength of the study is the purpose that was sought after. This manuscript focuses on an interesting topic and it has a high potential to contribute to the regulation of health behavior literature. The literature review is critically articulated with the general purpose (aim) of the study and previous research, and the bibliographic references are adjusted to the dimension of the subject under study. In general, the methodology was well applied, the results obtained are consistent, and the conclusion and limitations of the study were discussed and denote consistency. The sample size and the validation of the measures are the main limitations, such as the authors discuss, although there are other limitations related to the cross-sectional nature of the dataset, the sampling method selected, the characteristics of the sample, and the process of administering the measures, among other. Therefore, the results have to be taken with great caution.

In a revised version the authors should improve the paper in the following minor points related to methods, results and data interpretation.

The authors provide the psychometric properties of the measures in the current study… Why don't you provide the internal consistency of the subscales of the Self-Regulation vs External Regulation of Health, Psychological Reactance Scale and the Students Health Inventory?

I would like to have information on response rate.

Are you sure that the methodological design adopted was an ex post facto?

Regarding to the statistical analysis… What methods were used to estimate the structural models? Did you test that your data meets assumptions to perform these multivariate analyzes? A more detailed description and justification would be necessary. You checked univariate data distribution (i.e., Kolmogorov-Smirnoff test), but other pre-analyses components or preliminary analysis (e.g., linearity and atypical or outliers, missing and influential cases) should be provided, mainly the critical values of multivariate normality (e.g., Mardia’s coefficient or multivariate Kurtosis).

The authors used the Hoelter Index to determine sample size adequacy. However, the sample size should be backed up by an a priori power calculation based on the number of predictors, expected effect size, associated probability and levels of statistical power.

I think that the quality of the Figure 1 is not satisfactory.

Finally, I would suggest that you review English language and style. A re-read is necessary to remove errors and spelling errors (e.g., line 251 “Scale de psychological reactance”).

Author Response

Please, see the attach. Thanks
